# Defective Lysosomal Lipolysis Causes Prenatal Lipid Accumulation and Exacerbates Immediately after Birth

**DOI:** 10.3390/ijms221910416

**Published:** 2021-09-27

**Authors:** Katharina B. Kuentzel, Ivan Bradić, Alena Akhmetshina, Melanie Korbelius, Silvia Rainer, Dagmar Kolb, Martin Gauster, Nemanja Vujić, Dagmar Kratky

**Affiliations:** 1Gottfried Schatz Research Center, Molecular Biology and Biochemistry, Medical University of Graz, 8010 Graz, Austria; katharina.kuentzel@medunigraz.at (K.B.K.); ivan.bradic@medunigraz.at (I.B.); alena.akhmetshina@medunigraz.at (A.A.); m.korbelius@medunigraz.at (M.K.); silvia.rainer@medunigraz.at (S.R.); nemanja.vujic@medunigraz.at (N.V.); 2Gottfried Schatz Research Center, Cell Biology, Histology and Embryology, Medical University of Graz, 8010 Graz, Austria; dagmar.kolb@medunigraz.at (D.K.); martin.gauster@medunigraz.at (M.G.); 3Core Facility Ultrastructural Analysis, Medical University of Graz, 8010 Graz, Austria; 4BioTechMed-Graz, 8010 Graz, Austria

**Keywords:** cholesterol catabolism, lysosomal acid lipase, lysosomal storage disorder, placenta, development, mutant mouse models

## Abstract

Cholesterol and fatty acids are essential lipids that are critical for membrane biosynthesis and fetal organ development. Cholesteryl esters (CE) are degraded by hormone-sensitive lipase (HSL) in the cytosol and by lysosomal acid lipase (LAL) in the lysosome. Impaired LAL or HSL activity causes rare pathologies in humans, with HSL deficiency presenting less severe clinical manifestations. The infantile form of LAL deficiency, a lysosomal lipid storage disorder, leads to premature death. However, the importance of defective lysosomal CE degradation and its consequences during early life are incompletely understood. We therefore investigated how defective CE catabolism affects fetus and infant maturation using Lal and Hsl knockout (-/-) mouse models. This study demonstrates that defective lysosomal but not neutral lipolysis alters placental and fetal cholesterol homeostasis and exhibits an initial disease pathology already in utero as Lal-/- fetuses accumulate hepatic lysosomal lipids. Immediately after birth, LAL deficiency exacerbates with massive hepatic lysosomal lipid accumulation, which continues to worsen into young adulthood. Our data highlight the crucial role of LAL during early development, with the first weeks after birth being critical for aggravating LAL deficiency.

## 1. Introduction

Cholesterol and fatty acids (FA) are essential lipids for membrane biosynthesis and organ development. Two sources maintain an adequate supply of cholesterol: the body’s own (endogenous) production and exogenous intake from the diet in the postnatal period. Regardless of its source, cholesterol is transported in the bloodstream in form of lipoproteins. Unesterified free cholesterol (FC) is an important structural component of cell membranes to ensure membrane fluidity, a precursor for steroid hormones, bile acids, and vitamin D, and is involved in cell signaling processes. An excessive amount of FC is toxic and may disrupt cell membrane fluidity and integrity. Thus, excess cholesterol is stored in the cell in the inert form of cholesteryl esters (CE). CE are hydrolyzed in two distinct cellular compartments at different pH values by the action of specific enzymes. In the cytosol, hormone-sensitive lipase (HSL) releases FC and FA from CE as well as FA from diglycerides (DG), triglycerides (TG), monoglycerides, and retinyl esters at neutral pH [1,2]. In the acidic lysosomal lumen, lysosomal acid lipase (LAL) is the only enzyme known to hydrolyze CE, TG, DG, monoglycerides, and retinyl esters [3,4,5].

Impaired HSL or LAL activity causes rare pathologies in humans with different phenotypes and life expectancies. Patients affected by HSL deficiency appear lean but accumulate DG in their adipose tissue. Further progression of the disease includes insulin resistance, type 2 diabetes, and the development of fatty liver disease [6,7]. Hsl knockout (-/-) mice are lean, accumulate DG in adipose and muscle tissues, and show an anti-atherosclerotic lipoprotein profile [8,9]. Male Hsl-/- mice are sterile due to abnormalities in testis morphology and sperm development [10]. In contrast, LAL deficiency is a lysosomal storage disorder that results in a severe phenotype manifested by massive lysosomal lipid accumulation predominantly in the liver, intestine, adrenals, and macrophages. The early-onset form of this lysosomal storage disease, historically also called Wolman disease (WD), is characterized by <1% of residual enzymatic activity; affected infants die within the first year of life due to liver failure, malabsorption, and an inability to thrive [11]. The early deaths in WD patients underscore the importance of lysosomal lipid degradation during development. Patients with cholesteryl ester storage disease (CESD) retain up to 12% of residual LAL activity and may survive into adulthood, but suffer from hepatosplenomegaly, jaundice, gastrointestinal disturbances, and may develop multiorgan failure leading to death [11,12]. For more than 20 years, the Lal-/- mouse model has been used to study the progression and therapeutic options of this lysosomal storage disease [13,14,15,16,17]. However, the consequences of LAL deficiency during early development from the womb to young adulthood are incompletely understood.

As cholesterol plays an important role in fetal maturation and growth and can be delivered to the unborn via the placenta [18,19], we speculated that impaired cholesterol metabolism critically impacts their development. Therefore, we investigated the consequences of defective cholesterol catabolism in fetal and preadult mice. Our data demonstrate that HSL deficiency has little effect on placental and fetal development and does not interfere with placental lipid metabolism. However, the loss of LAL manifests itself in altered fetal cholesterol homeostasis at the end of pregnancy. Disease characteristics are already present 2 days after birth and continue to worsen during the first 4 weeks of life.

## 2. Results

### 2.1. Intracellular CE Hydrolases Are Expressed and Active in the Wild-Type (WT) Mouse Placenta

The mouse placenta, which we isolated on day 19 of pregnancy, is composed of three different layers, with the maternally derived decidua separated from the labyrinth by the junctional zone (Figure 1A). Lipids are heterogeneously distributed in the mouse placenta. We observed the most pronounced lipid staining in the labyrinth zone, where feto-maternal exchange occurs, whereas lipids are completely absent in the junctional zone (Figure 1A). Biochemical lipid estimation revealed that the mouse placenta contains more cholesterol, mainly FC, than TG (Figure 1B). We detected *Hsl* and *Lal* mRNA expression (Figure 1C), whereas other potential neutral CE hydrolases (i.e., *Nceh1* [20], *Ces1a*, *Ces1d* and *Ces2b* [21]) were not expressed in the WT mouse placenta (Figure 1C). Western blot analysis of tissue lysates confirmed the abundance of HSL and LAL protein in the WT placenta (Figure 1D). As mRNA and protein expression levels do not necessarily correlate with enzymatic activity due to posttranscriptional modifications [22], we determined acid and neutral CE hydrolase activities in the WT mouse placenta by measuring the release of FA from a ^14^C-labeled CE substrate. Adipose tissue, liver, and peritoneal macrophages were used as positive controls for neutral and acid CE hydrolase activities, whereas tissues from Hsl-/- and Lal-/- mice served as negative controls. As expected, neutral and acid CE hydrolases were less active in the placenta than in control tissues but comparable to macrophages (Figure 1E,F). These results demonstrated that HSL and LAL are expressed and active in the mouse placenta.

### 2.2. Defective Neutral CE Hydrolysis Has Minor Impact on Placental and Fetal Lipid Metabolism

We used Hsl-/- mice (Appendix A) as a model to study the impact of neutral CE hydrolysis on mouse placental lipid metabolism. Isolated placentas from Hsl-/- mice had a similar macroscopic appearance (data not shown) and weight as WT controls (Figure 2A). In addition, fetal weights of Hsl-/- offspring were unaltered (Figure 2A), and WT and Hsl-/- mice had comparable placental efficiency (Appendix A), producing pups of similar size. As expected from these results, fetal organ weights (liver and brain) were unchanged (Figure 2B), indicating that HSL deficiency did neither affect placental or fetal growth nor organ development in utero.

Despite a 35% decrease in neutral CE hydrolase activity in Hsl-/- placentas (Figure 2C), neutral lipid staining of placental sections with ORO (Figure 2D) and placental lipid concentrations (Figure 2E) were comparable between WT and Hsl-/- mice. The absence of other potential neutral CE hydrolases (*Nceh1*, *Ces1a*, *Ces1d*, *Ces2b*) (Figure 1C) together with unchanged *Lal* mRNA expression (Figure 2F) and LAL activity (Figure 2G) suggested that the unchanged CE concentrations are not the result of upregulated acid CE hydrolysis to compensate for reduced neutral CE hydrolysis. However, mRNA expression of genes regulating cholesterol synthesis (*Srebp2* and *Hmgcr*) was downregulated by more than 50% in Hsl-/- placentas (Figure 2H), arguing for reduced cholesterol biosynthesis. Neutral and acid TG hydrolase activities remained unchanged between the genotypes (Figure 2I).

Hsl-/- mice and HSL-deficient humans preferentially accumulate neutral lipids (mainly DG) in adipose tissue [6,8] and have increased liver fat concentrations [8,23]. We therefore measured fetal hepatic lipid concentrations. Similar to placental tissue, Hsl-/- offspring did not accumulate neutral lipids in their livers (Appendix A). These findings suggest that loss of HSL as neutral CE and TG hydrolase fails to impact lipid metabolism in the mouse placenta and fetus.

### 2.3. LAL Deficiency Alters Placental Lipid Homeostasis

Apart from cytosolic lipid droplets, CE are stored in lysosomes and degraded by LAL at acidic pH. LAL is highly expressed at mRNA and protein levels in the mouse placenta (Figure 1C,D). Therefore, Lal-/- mice served as a model to study the impact of defective lysosomal CE hydrolysis on the placenta and pregnancy outcome. We confirmed the loss of Lal in the placenta by gene expression analysis, whereas the expression of other intracellular lipases (adipose triglyceride lipase (*Atgl*) and *Hsl*) was unaltered (Figure 3A). In line, we measured drastically reduced acid CE hydrolase activity in the placentas from Lal-/- mice but not in the decidua (Figure 3B) due to its heterozygous (Lal+/-) maternal origin.

Placental weight and efficiency were not influenced by the loss of LAL (Figure 3C). Consistent with reduced acid CE hydrolase activity, we observed more neutral lipids by ORO staining (Figure 3D) and elevated CE concentrations in Lal-/- placentas, whereas TG levels remained unchanged (Figure 3E). In livers of adult Lal-/- mice [17] and LAL-deficient patients [24], the increased amount of neutral lipids (CE and TG) accumulates in lysosomes and can form CE crystals. These thin, crystalline structures were also visible in electron micrographs of Lal-/- placentas (Figure 3F, black arrows). As CE are mainly “trapped” in lysosomes of affected patients and adult Lal-/- mice, we double stained histologic sections from the placenta with ORO and the lysosomal membrane marker lysosome associated membrane protein 1 (LAMP1). Fluorescence microscopy revealed the abundance of cytosolic lipids and lysosomes in WT placenta (Figure 3G). In Lal-/- placentas, LAMP1 immunostaining revealed an increased number of lysosomes. Although lysosomal lipid accumulation was evident in Lal-/- placentas (Figure 3F, white arrows), most of the lipids were located outside the lysosome. Thus, we determined neutral TG hydrolase activity as a measure of ATGL activity, which was slightly but significantly lower in Lal-/- compared to WT placentas (Figure 3H). These results indicated that LAL deficiency alters placental lipid homeostasis, leading to CE accumulation, CE crystal formation, and slightly reduced neutral TG hydrolase activity.

### 2.4. LAL Deficiency Manifests Itself Already in the Fetus

As the Lal-/- placenta displayed some characteristic features of LAL deficiency such as CE accumulation and formation of small CE crystals, we speculated that cholesterol metabolism might also be altered in the unborn Lal-/- offspring. Adult mice globally lacking LAL are smaller and lighter compared to their WT littermates [17]. Therefore, we investigated whether the unborn pups suffer from intrauterine growth restriction (calculated by a brain-to-liver ratio > 3, [25]). However, on day 19 of pregnancy, fetal weight was comparable between the two genotypes (Figure 4A). Unaltered fetal brain and liver weight, and a brain-to-liver ratio < 3 (Appendix A) suggested no fetal growth restriction in Lal-/- offspring.

Although Lal-/- and WT offspring appeared macroscopically similar and ORO staining of whole fetal sections showed no differences in the abundance of neutral lipids (Appendix A), we found increased CE concentrations in total body lysates from Lal-/- compared to control fetuses (Figure 4B). In whole-body Lal-/- mice, CE and TG are mainly trapped in lysosomes of the liver [17], in macrophages, and in the small intestine [16]. We therefore isolated liver and intestine from Lal-/- fetuses for cryosectioning and lipid quantification. ORO-stained sections from fetal Lal-/- livers revealed a slightly increased abundance of neutral lipids (Figure 4C), which was associated with elevated hepatic CE but not TG concentrations (Figure 4D). We observed lipid-filled lysosomes in electron micrographs of livers from Lal-/- offspring (Figure 4E and Appendix A), which was confirmed by immunofluorescent double staining with LAMP1 and ORO (Appendix A). Both methods additionally revealed remaining cytosolic lipid droplets in Lal-/- fetal livers (Figure 4E and Appendix A). Similar to the placenta, we also found CE crystals (Figure 4E and Appendix A). Neutral lipid staining with ORO showed a higher amount of lipids in the small intestine (Figure 4F), which was attributable to increased CE concentrations (Figure 4G). These results suggested that unborn Lal-/- offspring are already affected by LAL deficiency.

### 2.5. Rapid Disease Progression of LAL Deficiency Shortly after Birth

Our data so far have shown that cholesterol homeostasis is altered in Lal-/- offspring in utero and that they exhibit an initial manifestation of LAL deficiency. These results prompted us to next investigate disease progression during the first weeks of life. Therefore, we chose three time points representing important stages in early life: after birth (2 days), during lactation (2 weeks), and after weaning (4 weeks after birth).

As early as 2 days after birth, Lal-/- mice displayed slightly reduced body weight, whereas liver and intestinal weights remained comparable to their WT littermates (Figure 5A). Despite their unaltered size, Lal-/- livers appeared macroscopically paler and more yellowish than the WT tissues (Figure 5B). Lipid staining confirmed this macroscopic change with a massive accumulation of neutral lipids (Figure 5C) that was due to increases in both TG and CE concentrations in Lal-/- livers (Figure 5D). Contrary to the liver, neutral lipids in the intestine of WT and Lal-/- offspring remained comparable (Figure 5E). These data suggest that Lal-/- mice are already slightly smaller immediately after birth and suffer from fatty liver.

### 2.6. Aggravation of LAL Deficiency 4 Weeks after Birth

To further follow disease progression early in life, we sacrificed WT and Lal-/- pups 2 and 4 weeks postpartum. The body weight of Lal-/- mice was significantly lower than that of their WT littermates at both time points (Appendix A and Figure 6A). Together with increased liver weight and an ~50% reduction of subcutaneous white adipose tissue (sWAT) at the age of 2 and 4 weeks, the adult phenotype of Lal-/- mice was fully developed (Appendix A and Figure 6A). Lipid staining revealed a pronounced lipid accumulation in Lal-/- livers, regardless of their age (Appendix A and Figure 6B). Interestingly, the lipid composition shifted from 3.6-fold TG and 3.1-fold CE accumulation at 2 weeks (Appendix A) to a 25-fold increase in CE concentrations at 4 weeks of life (Figure 6C). Lipids entrapped in lysosomes (Figure 6D) increased from 2 to 4 weeks after birth (Appendix A). In the lysosomes of Lal-/- livers, we identified linear structures as CE crystals (Figure 6D and Appendix A (enlarged)). Livers of adult Lal-/- mice completely lack cytosolic lipid droplets, in line with almost complete loss of WAT as endogenous lipid stores [17]. In contrast to the liver, intestinal lipid content at 2 weeks of age was similar between the genotypes (Appendix A) but showed markedly increased TG and CE concentrations at 4 weeks of age (Figure 6E).

Taken together, our data demonstrate that LAL deficiency exhibits its major phenotypic features already in utero. With dietary fat intake through the maternal milk as the main source of energy, the disease progresses rapidly in Lal-/- pups, with full manifestation of LAL deficiency as early as 4 weeks after birth.

## 3. Discussion

During fetal development, cholesterol in the fetal tissue is either derived from endogenous synthesis or supplied from the maternal side [18]. The majority of cholesterol in the circulation is transported in the form of CE within lipoproteins, which are taken up by receptor-mediated endocytosis [26]. Within the cell, CE can be stored in cytosolic lipid droplets and lysosomes and catabolized by intracellular CE hydrolases.

Our results demonstrate that HSL plays only a marginal role as neutral TG and CE hydrolase in the mouse placenta and unborn pups as placental and fetal lipid concentrations remained unchanged in Hsl-/- mice. HSL exhibits a broad substrate specificity to hydrolyze DG, CE, retinyl esters, monoglycerides, and TG (but with low hydrolytic activity) [1,2,27]. Unaltered TG levels in Hsl-/- placentas are consistent with data from Hsl-/- adipose tissue [8]. mRNA expression and enzymatic activity of ATGL, which catalyzes the first step in neutral TG hydrolysis [28], were unaltered upon the loss of HSL. Despite reduced neutral CE hydrolase activity, we failed to observe changes in placental CE concentrations. As *Nceh1*, *Ces1a*, *Ces1d*, *and Ces2b* as potential CE hydrolases were not expressed in the mouse placenta, the only remaining candidate that could compensate for the loss of HSL in the mouse placenta is LAL. Considering comparable *Lal* expression and unchanged acid CE hydrolase activity in Hsl-/- placentas, the only plausible explanation for the unchanged CE content may be reduced endogenous cholesterol synthesis. Indeed, we observed downregulation of cholesterol synthesis-related genes in the placentas of Hsl-/- animals, suggesting a compensatory decrease in cholesterol production.

Of note, this study provides evidence that LAL deficiency originates already in utero and becomes severely exacerbated during the first 4 weeks of life, at least in mice. LAL deficiency is a rare autosomal recessive lysosomal storage disorder characterized by progressive accumulation of neutral lipids in multiple cells and organs [12]. Previous studies have focused only on the phenotype of adult Lal-/- mice [15,16,17,29] and have shown that Lal-/- mice appear normal at birth and have no altered phenotype by 1.5 months of age on a mixed 129Sv/CF-1 background [13,14,16]. Data on LAL deficiency in utero and from birth to juvenility were lacking. In this study, we shed new light on several developmental stages from the late fetal to the early adult phase.

In contrast to HSL, the loss of LAL alters placental and fetal cholesterol homeostasis already at the end of pregnancy. Using electron microscopy, we observed CE crystals in the placenta of Lal-/- mice. These crystals are known to be present in livers of adult Lal-/- mice [13,14,17] and LAL-deficient patients [24]. Although the presence of crystallized FC has been described in atherosclerotic lesions of humans and rabbits [30,31,32,33], the existence of FC crystals in Lal-/- mice is unlikely because FC concentrations in placentas and fetal livers were comparable to levels in WT tissues. Another indication in favor of CE crystals is their lysosomal location. Because of their sharp structure, they might also be found in the cytosol when they rupture the lysosomes, as described for FC crystals in atherosclerotic lesions [34]. In contrast to the thin and small placental CE crystals, the fetal liver crystals appear larger and resemble the crystal structures found in adult mice. One factor contributing to the larger fetal crystal size could be the higher CE concentration in the fetal liver than in the placenta of Lal-/- mice.

In contrast to our expectations, the loss of LAL did not result in a massive lysosomal lipid accumulation in the placenta. The increased ORO staining of placental cytosolic lipid droplets in Lal-/- compared to WT placenta may be due to more but smaller lipid droplets and reduced ATGL activity. Several previous reports indicate a reduction of the overall intracellular lipolytic machinery upon genetic loss of a neutral or acid lipase [35,36]. Moreover, the main substrate for placental and fetal growth is glucose [37,38], whereas FA taken up by diffusion or FA transport proteins are used to a lesser extent. In addition, lipoprotein lipase is actively involved in the hydrolysis of TG in lipoproteins and the release of FA, thereby limiting the need for lipoprotein uptake by receptor-mediated endocytosis and the incorporation of neutral lipids into placental lysosomes. Lal-/- mice are poor breeders, and heterozygous dams bred in this study still express active LAL enzyme that could be transported to the placenta via the bloodstream. To date, the secretion and reuptake of LAL are not fully understood, but e.g., macrophages express [39] and secrete LAL [40,41]. Whether LAL secretion from placental Lal+/- macrophages plays a role in the observed phenotype in utero requires further investigation.

In contrast to previous reports that Lal-/- mice appear normal at birth [13,16], Lal-/- fetuses on the C57BL/6J background were affected by LAL deficiency already at the end of gestation. Our results are consistent with a report of human LAL deficiency in a miscarriage, in which the pathologic liver phenotype was already detectable in week 17 of gestation [42] and a second case of prenatal hepatosplenomegaly in gestational week 34 [43]. Immediately after birth, Lal-/- offspring accumulated hepatic TG and CE, which could even be observed macroscopically. Despite unaltered TG in fetuses and the placentas, TG concentrations were drastically increased in the livers of 2-day-old Lal-/- offspring. This shift in hepatic lipid accumulation could be due to dietary changes, as mouse milk contains a large amount of fat and is essential for pup survival and development, being the first and only source of nutrition directly after birth [44,45,46]. In contrast to the liver, the intestine did not appear to be affected by Lal deficiency 2 days and 2 weeks postpartum. Interestingly, the switch to standard chow diet increased intestinal CE and TG levels in Lal-/- mice. As lipid accumulation in the intestine of adult Lal-/- mice is probably caused mainly by macrophages (Bianco et al., unpublished observation), it is likely that lipid-filled macrophages infiltrate the small intestine only 4 weeks after birth and not before. Another speculation may be that the intestine of Lal-/- pups still receive active LAL from the heterozygous mother; however, this needs to be addressed in a further study.

Taken together, our data underscore the crucial role of LAL during early development, as the Lal-/- fetus is already affected by hepatic CE accumulation at the end of gestation. In addition to intrauterine development, the first weeks after birth seem to be critical for the aggravation of LAL deficiency.

## 4. Materials and Methods

### 4.1. Animals and Sample Collection

LAL knockout (Lal-/-) mice [13] were bred heterozygously (Lal+/− × Lal+/−) and genotyped by PCR after isolation of genomic DNA from ear clipping. HSL knockout (Hsl-/-) animals on the C57BL/6J background [8] were generated by crossing Hsl-/- females with Hsl+/− males due to sterility of homozygous males [10]. All animals were housed under standard laboratory conditions in a 12 h light/dark cycle and in a clean, temperature controlled (22 ± 1 °C) environment. Mice had unlimited access to standard chow diet (1324, Altromin, Lage, Germany). Breeding cages contained one male and two female mice at the age of 8–12 weeks. Four days after the start of breeding, the male was removed, and pregnancy was confirmed based on the weight gain between days 4 and 12 [47]. On day 19 of pregnancy, dams were sacrificed by cervical dislocation and tissues were collected. The decidua was removed after weighing all placentas to avoid contamination from maternal tissue. Fetuses were genotyped after isolation of genomic DNA from tail tips by PCR. Litters with less than 5 animals were excluded from all experiments and analyses. All experiments were performed with at least 3 samples from different dams/pregnancies. Young animals were sacrificed between 2 days and 4 weeks after birth. All experiments were performed in accordance with the European Directive 2010/63/EU and approved by the Austrian Federal Ministry of Education, Science and Research (Vienna, Austria; BMWFW-66.010/165-V/3b/2019 and BMBWF-66.010/0197-V/3b/2017).

### 4.2. RNA Isolation, cDNA Preparation and Real-Time PCR

Total RNA was isolated with TRIsure^TM^ reagent according to the manufacturer’s protocol (Meridian Bioscience, Cincinnati, OH, USA). Two micrograms of RNA were reverse transcribed using the High Capacity cDNA Reverse Transcription Kit (Applied Biosciences, Carlsbad, CA, USA) and 6 ng of cDNA were used for quantitative real-time PCR. All samples were analyzed in duplicate and normalized to *cyclophilin A* as housekeeping gene. Primers are listed in Table 1.

### 4.3. Western Blot

Tissue samples were sonicated twice for 10 s on ice in RIPA buffer (150 mM sodium chloride (Carl Roth, Karlsruhe, Germany), 1% NP-40 (Sigma-Aldrich, St. Louis, MO, USA), 0.5% sodium deoxycholate (Sigma-Aldrich, St.Louis, MO, USA), 0.1% SDS (Carl Roth, Karlsruhe, Germany), 50 mM Tris (Carl Roth, Karlsruhe, Germany), pH 8). Protein concentrations were determined in the supernatant after centrifugation for 10 min at 1000× *g* and 4 °C using the DC™ Protein Assay Kit (Bio-Rad Laboratories, Hercules, CA, USA). Fifty micrograms of protein were separated by SDS-PAGE and transferred to a nitrocellulose membrane. The following anti-rabbit antibodies were used: LAL (TA 309730, 1:1000, OriGene Technologies, Rockville, MD, USA) and HSL (#4107, 1:800, Cell Signaling Technologies, Danvers, MA, USA). Monoclonal anti-mouse ß-Actin (sc-47778, 1:10,000, Santa Cruz, Heidelberg, Germany) was used as a loading control. White adipose and liver tissue from WT mice served as controls for the expression of neutral and acid lipases, respectively.

### 4.4. Immunofluorescence and ORO Staining

Paraformaldehyde (VWR, International, Radnor, PA, USA) -fixed tissue was transferred to 30% sucrose (Carl Roth, Karlsruhe, Germany) solution prior to cryosectioning. Sections (5 µm) were cut, rehydrated in PBS, and blocked with 0.05% PBST (0.05% Tween-20 (VWR International, Radnor, PA, USA) in PBS) containing 10% goat serum (Szabo-Scandic, Vienna, Austria) for 30 min. Sections were incubated with anti-LAMP1 antibody (1D4B, 1:25, Developmental Studies Hybridoma Bank, Iowa City, IA, USA) overnight at 4 °C, washed with PBS, and incubated with anti-rat Alexa Fluor^®^ 488 secondary antibody (1:250, ThermoFisher Scientific, Waltham, MA, USA) for 2 h at room temperature. For co-staining with ORO (Sigma-Aldrich, St. Louis, MO, USA), slides were incubated for 1 h in the freshly prepared ORO staining solution and subsequently stained for 10 min with DAPI (Sigma-Aldrich, St. Louis, MO, USA) to visualize nuclei. Slides were mounted using Dako Fluorescence Mounting Medium (Dako North America Inc., Carpinteria, CA, USA) and visualized on an Olympus BX63 microscope equipped with an Olympus DP73 camera (Olympus, Shinjuku, Japan).

### 4.5. Lipid Extraction

Tissues were sonicated twice for 10 s on ice in lysis buffer (100 mM potassium phosphate (Carl Roth, Karlsruhe, Germany), 250 mM sucrose (Carl Roth, Karlsruhe, Germany), 1 mM EDTA (ThermoFisher Scientific, Waltham, MA, USA), pH 7) and the protein amount was determined after centrifugation as described above. Lipids were extracted from 1 mg protein following Folch’s method [48]. TG, TC, and FC concentrations were determined using enzymatic kits following the manufacturer’s guidelines (DiaSys, Holzheim, Germany). CE concentrations were calculated by subtraction of FC from TC values. All lipid concentrations were normalized to protein content.

### 4.6. CE and TG Hydrolase Activity Assays

To determine acid CE and TG hydrolase activities, tissues were lysed in acid citrate buffer (containing 54% of 100 mM citric acid monohydrate and 46% of 100 mM trisodium citrate, dehydrated, pH 4.2; (Carl Roth, Karlsruhe, Germany)) or neutral lysis buffer (1 mM DTT (Carl Roth, Karlsruhe, Germany), pH 7), sonicated twice for 10 s on ice, and centrifuged at 1000× *g* and 4 °C for 10 min. Protein concentrations were determined in the supernatant. Thereafter, 50 μg of protein were diluted to a final volume of 100 μL in citrate or neutral lysis buffer. We used 0.2 mM cholesteryl oleate/sample, 0.04 μCi/sample cholesteryl [1-^14^C]-oleate (Amersham Biosciences, Piscataway, NJ, USA) and 35.5 μg mixed micelles of phosphatidylcholine and phosphatidylinositol (3:1) as substrate to determine CE hydrolase activity. The substrate for the TG hydrolase activity assay contained 0.3 mM triolein/sample, 0.5 µCi/sample [9,10-^3^H(N)]-triolein (Perkin Elmer, Waltham, MA, USA), and 3.5 µg of above-mentioned mixed micelles. Each substrate contained FA-free BSA (Biowest, Nuaillé, France) at a final concentration of 2% in 100 mM citrate or phosphate buffer, respectively. Liver tissues from WT and Lal-/- mice (for acid CE and TG hydrolase activity) and white adipose tissue from WT and Atgl-/- (neutral TG hydrolase activity) or Hsl-/- (neutral CE hydrolase activity) mice were used as positive and negative controls, respectively. Samples were incubated for 1 h with the respective substrate at 37 °C and the reaction was terminated by the addition of 3.25 mL stop solution (methanol:choroform:n-heptane, 10:9:7, v:v:v; all Carl Roth, Karlsruhe, Germany) and 1 mL of 100 mM potassium carbonate (pH 10.5), (Carl Roth, Karlsruhe, Germany). After vortexing and centrifugation at 3220× *g* and 4 °C for 15 min, the radioactivity in 1 mL of the upper phase was determined by liquid scintillation counting and the release of FA was calculated as previously described [22].

### 4.7. Electron Microscopy

Freshly harvested placentas and fetal hepatic tissue were immediately fixed in 2.5% (w:v) glutaraldehyde (Electron Microscopy Science, Hatfield, PA, USA) and 2% (w:v) paraformaldehyde (Merck KGaA, Darmstadt, Germany), buffered in 100 mM cacodylate buffer (Merck KGaA, Darmstadt, Germany), pH 7.4, incubated at room temperature for 3 h, and post-fixed in 2% osmium tetroxide (Electron Microscopy Science, Hatfield, PA, USA) (diluted in 200 mM cacodylate buffer) for 2–3 h at room temperature. After washing for 2 h in 100 mM cacodylate buffer, the specimens were dehydrated in a graded series of ethanol (50%, 70%, 80%, 96%, 100% p.a.) and stepwise infiltrated with propylene oxide/embedding resin (Agar Scientific, Essex, Great Britain): Propylene oxide (Fluka, Sigma Aldrich GmbH, Germany) for 1 h at room temperature, followed by propylene oxide/resin 1:1 for 3 h at room temperature, propylene oxide/resin 1:3 overnight at 4 °C, embedding in pure resin (TAAB Laboratories Equipment Ltd, England, UK), and polymerization (2 × 1.5 h at 48 °C). Ultrathin sections (70 nm) were stained with lead citrate (Leica Microsystems, Wetzlar, Germany) and platinum blue (IBI Labs Inc., Boca Raton, FL, USA) and examined on a Tecnai G2 electron microscope (ThermoFisher Scientific, Waltham, MA, USA) equipped with a Gatan Ultrascan 1000 CD Camera.

### 4.8. Statistical Analysis

Statistical analysis was performed using GraphPad Prism software 5.01 (GraphPad Software Inc., San Diego, CA, USA). Significance was calculated by unpaired Student’s *t*-test. Data are shown as mean + SD. For real-time PCR analysis, the 2^−ΔΔCT^ method was used. The following significance levels were used: * *p* < 0.05, ** *p* ≤ 0.01, *** *p* ≤ 0.001.

## Figures and Tables

**Figure 1 ijms-22-10416-f001:**
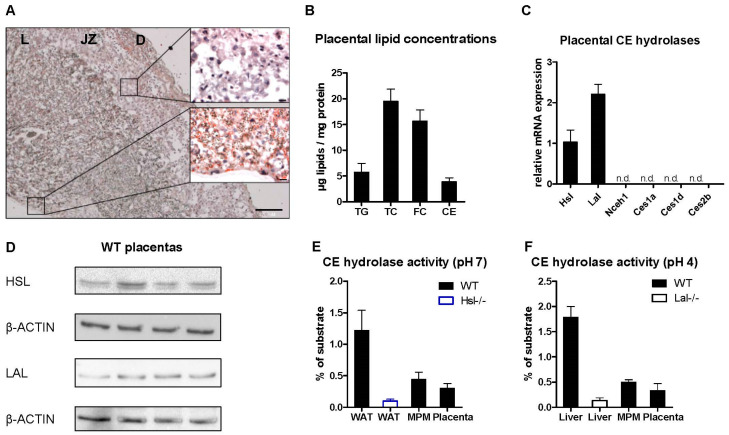
Intracellular cholesteryl ester (CE)-hydrolyzing enzymes are expressed and active in wild type (WT) mouse placenta. Placentas were isolated from WT mice on day 19 of pregnancy. (**A**) ORO staining of a placental section; decidua (D), junctional zone (JZ), and labyrinth (L). (**B**) Triglyceride (TG), total cholesterol (TC), free cholesterol (FC), and CE concentrations in the mouse placenta. Magnification, 4×; Scale bar, 200 µm. (**C**) mRNA expression of CE hydrolases in mouse WT placenta relative to the expression of *cyclophilin A* as a housekeeping gene. (**D**) Hormone-sensitive lipase (HSL) and lysosomal acid lipase (LAL) protein expression in the placenta. (**E**) Neutral and (**F**) acid CE hydrolase activities in the placenta. White adipose tissue (WAT), liver, and mouse peritoneal macrophages (MPM) were assayed as controls. Data represent mean values + SD (*n* = 4 placentas from different dams and tissues/cells from different mice).

**Figure 2 ijms-22-10416-f002:**
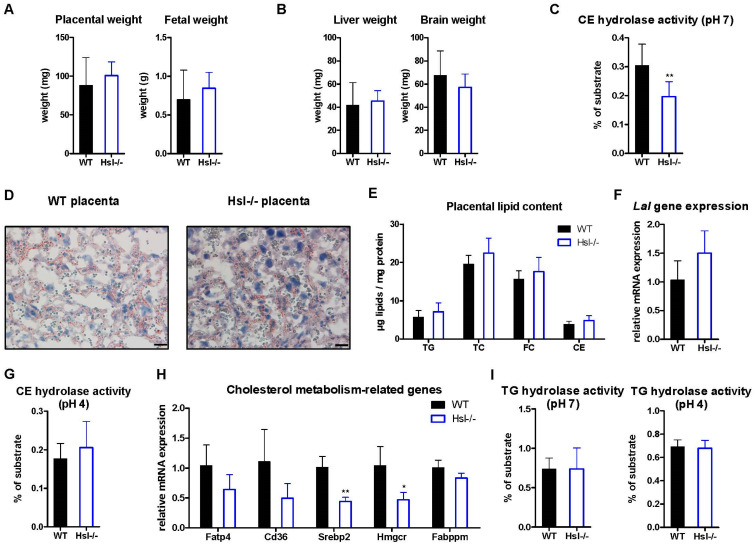
Hormone-sensitive lipase (HSL) deficiency does neither affect mouse placental or fetal weight nor lipid content. Placentas were isolated from WT and Hsl-/- mice on day 19 of pregnancy. (**A**) Placental and fetal weight (*n* = 36–41 WT, 24 Hsl-/-) as well as (**B**) fetal organ weights (*n* = 18 WT, 22 Hsl-/-). (**C**) Neutral CE hydrolase activity in the placenta (*n* = 10). (**D**) ORO staining of cryosections from placentas. Magnification, 40×; Scale bar, 20 µm. (**E**) Placental lipid concentrations (*n* = 4). (**F**) Gene expression of placental *Lal* (*n* = 4). (**G**) Acid CE hydrolase activity in the placenta (*n* = 10). (**H**) mRNA expression of placental genes related to cholesterol metabolism. (**I**) Neutral and acid TG hydrolase activities in placentas (*n* = 10). Data represent mean values + SD from different dams. Statistically significant differences were calculated by two-tailed Student’s *t*-test; * *p* < 0.05, ** *p* ≤ 0.01.

**Figure 3 ijms-22-10416-f003:**
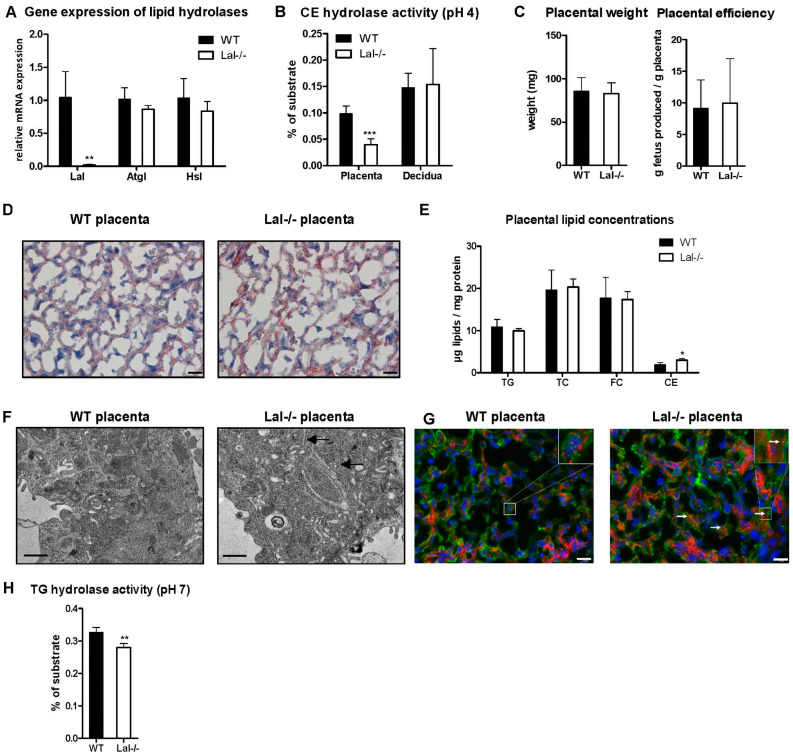
Lysosomal acid lipase (LAL) deficiency alters placental lipid homeostasis in utero. Placentas were isolated from WT and Lal-/- mice on day 19 of pregnancy. (**A**) Gene expression of lipases in the placenta. (**B**) Acid CE hydrolase activity in placenta and decidua. (**C**) Placental weight and efficiency (*n* = 26 WT, 34 Lal-/-). (**D**) ORO staining of cryosections from placental tissue. Magnification, 40×; Scale bar, 20 µm. (**E**) Placental lipid concentrations. (**F**) Electron micrographs of WT and Lal-/- placenta; black arrows indicate CE crystals. Scale bar, 1 µm. (**G**) Immunofluorescence double staining for neutral lipids (ORO, red), lysosomes (LAMP1, green), and nuclei (DAPI, blue) in the placenta; white arrows indicate lipids co-localizing with lysosomes. Magnification, 40×; scale bar, 20 µm. (**H**) Neutral TG hydrolase activity. Data represent mean values + SD from different dams (*n* = 4). Statistically significant differences were calculated by two-tailed Student’s *t*-test; * *p* < 0.05, ** *p* ≤ 0.01, *** *p* ≤ 0.001.

**Figure 4 ijms-22-10416-f004:**
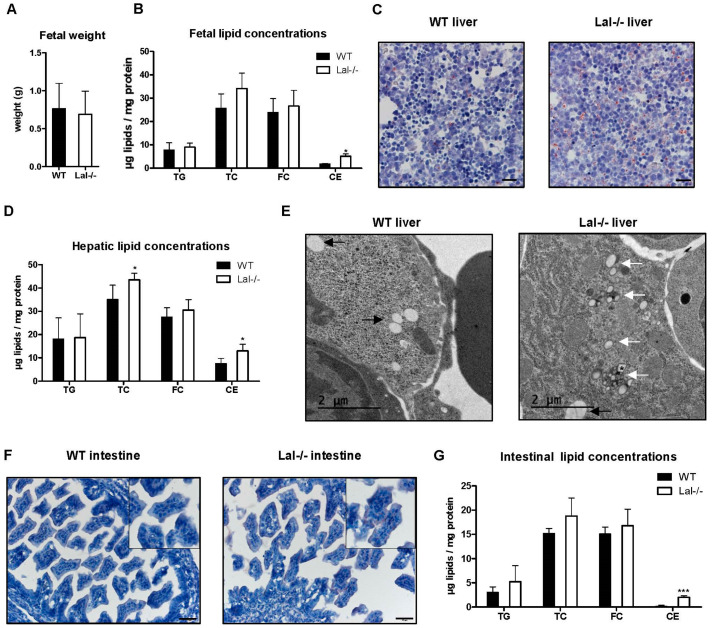
CE but not TG accumulation in fetal liver and intestine of Lal-/- mice. Fetuses were isolated from WT and Lal-/- mice on day 19 of pregnancy. (**A**) Fetal weight (*n* = 24 WT, 34 Lal-/-) and (**B**) lipid concentrations (*n* = 3 WT, 4 Lal-/-). (**C**) ORO staining of cryosections from fetal liver. Magnification, 40×; Scale bar, 20 µm. (**D**) Fetal hepatic lipid levels (*n* = 4). (**E**) Electron micrographs of fetal liver; white arrows indicate lipid-filled lysosomes, black arrows indicate cytosolic lipid, * indicates CE crystal. (**F**) ORO staining of cryosections from fetal intestine. Magnification, 20×; Scale bar, 50 µm. (**G**) Fetal intestinal lipid levels (*n* = 3–5). Data represent mean values + SD. Statistically significant differences were calculated by two-tailed Student’s *t*-test; * *p* < 0.05, *** *p* ≤ 0.001.

**Figure 5 ijms-22-10416-f005:**
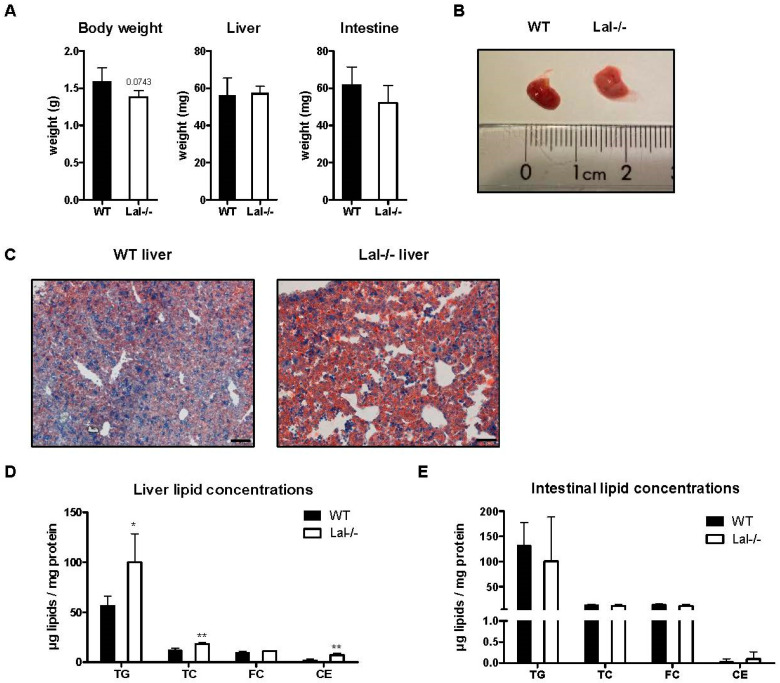
Two-day-old Lal-/- mice accumulate hepatic lipids. Intestinal and hepatic tissues were isolated from WT and Lal-/- mice 2 days after birth. (**A**) Body and organ weights (*n* = 6 WT, 4 Lal-/-). (**B**) Macroscopic and (**C**) histologic analysis of WT and Lal-/- livers. Magnification, 20x; Scale bar, 50 µm. (**D**) Biochemical quantification of hepatic and (**E**) intestinal lipid concentrations (*n* = 4 WT, 3 Lal-/-). Data represent mean values + SD. Statistically significant differences were calculated by two-tailed Student’s *t*-test; * *p* < 0.05, ** *p* ≤ 0.01.

**Figure 6 ijms-22-10416-f006:**
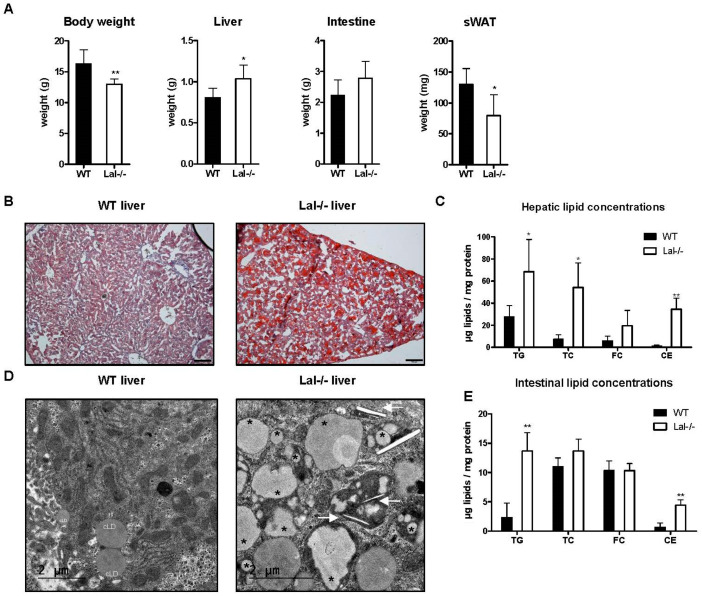
Aggravation of LAL deficiency during the first 4 weeks of life. Organs were isolated from WT and Lal-/- mice 4 weeks after birth. (**A**) Body and organ weights and (**B**) ORO staining of liver cryosections. Magnification, 10x; Scale bar, 100 µm. (**C**) Hepatic lipid concentrations (*n* = 4 WT, 5 Lal-/-). (**D**) Electron micrographs of WT and Lal-/- livers; cLD, cytosolic lipid droplet; white arrows indicate lysosomal CE crystals, * indicate lysosomal lipids. Scale bar, 2 µm. (**E**) Intestinal lipid concentrations (*n* = 4). Data represent mean values + SD. Statistically significant differences were calculated by two-tailed Student’s *t*-test: * *p* < 0.05, ** *p* ≤ 0.01.

**Table 1 ijms-22-10416-t001:** Primers for real-time PCR.

Gene	Forward Sequence 5′–3′	Reverse Sequence 5′–3′
*Atgl*	GCCACTCACATCTACGGAGC	GACAGCCACGGATGGTGTTC
*Lal*	GCTGGCTTTGATGTGTGGATG	ATGGTGCAGCCTTGAGAATGA
*Hsl*	GATTTACGCACGATGACACAGT	ACCTGCAAAGACATTAGACAGC
*Nceh1*	TCGCAGCGGCTCTTCTGGTT	GATGCTGCTGGACGCCACTT
*Ces1a*	TATGTGCTCGCAAATTACAGGAG	CACCAGAGAGTAAACCGCCTC
*Ces1d*	ATGGAGGTGGACTGGTGGTG	AGTGCAGCCACCTGGTCCAA
*Ces2b*	AACGATGAGTTTGGTTGGACC	GAGGCAGCATCAGTTGTGC
*Srebp2*	TGAAGGACTTAGTCATGGGCAC	CGCAGCTTGTGATTGACCT
*Hmgcr*	TGTTCACCGGCAACAACAAGA	CCGCGTTATCGTCAGGATGA
*Cd36*	GCAGGTCTATCTACGCTGTG	GGTTGTCTGGATTCTGGAGG
*Fatppm*	GGACCTCCAGATCCCATCCT	GGTTTTCCGTTATCATCCCGGTA
*Fatp4*	GGCACAGACACTCACTGGAC	TGCGGTTTTCCATAAAGAGGG
*CyclophilinA*	GAGCTGTTTGCAGACAAAGTTC	CCCTGGCACATGAATCCTGG

## Data Availability

The data presented in this study are available upon request from the corresponding author.

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
