# Peer review of "Defective Lysosomal Lipolysis Causes Prenatal Lipid Accumulation and Exacerbates Immediately after Birth"

_ijms, 2021, doi:10.3390/ijms221910416_

Round 1

Reviewer 1 Report

It was a great treat to review this manuscript. It is an interesting subject and written in an engaging logical style. The experimental design, figures and descriptions are excellent. I have two very minor questions: 1) line 187 - is 'In global' the right choice of words ('In general' maybe?), 2) ATGL on line 268 has no full name and I wondered what it was abbreviated from - I cannot find the full name but maybe I missed it. 

Author Response

Response to Reviewer 1:

We would like to thank the Reviewer for reviewing our manuscript and the positive response which helped to improve the quality of our manuscript. We have addressed the comments and suggestions according to the Reviewer’s comments as follows:

1) line 187 - is 'In global' the right choice of words ('In general' maybe?),

Answer: In “global Lal-/- mice” means that these mice are whole-body Lal-deficient mice. “In general” would be a wrong wording. To avoid misunderstandings, we have changed it to “whole-body Lal-/- mice” in the revised version of the manuscript.

2) ATGL on line 268 has no full name and I wondered what it was abbreviated from - I cannot find the full name but maybe I missed it.

Answer: We apologize that we missed to mention the full name. Adipose triglyceride lipase (ATGL) is the rate-limiting enzyme in cytosolic lipid hydrolysis by mediating FA release from triglycerides. We have included the full name in lane 144 when it is mentioned the first time.

Reviewer 2 Report

I really appreciate this nice experimental work, which is well conducted and well written. I congratulate the authors for this good work, and I just would like to highlight a couple of little inconsistencies:

- Figure 4: letter for panel "A", "B", "C", and "F" are missing. Notably, Figure 4E is indicated as electron micrographs but it seems a light microscopy image to me. Please provide some explanations.

- Figure S3C: please provide other magnification and higher resolution for this panel, since lipid-filled lysosomes are not clearly visible

Author Response

Response to Reviewer 2:

We would like to thank the Reviewer for reviewing our manuscript and the positive response which helped to improve the quality of our manuscript. We have addressed the comments and suggestions according to the Reviewer’s comments, as follows:

- Figure 4: letter for panel "A", "B", "C", and "F" are missing. Notably, Figure 4E is indicated as electron micrographs but it seems a light microscopy image to me. Please provide some explanations.

Answer: We have included A, B, C, and F, which somehow got lost. Figure 4E is an electron micrograph; it can be distinguished from a light microscopy image by the higher magnification and visibility of single lipid droplets and other cell organelles like lysosomes in the case of Lal-/- sections. Furthermore, the grey scale of the electron micrograph is different from the stained light microscopy images.

- Figure S3C: please provide other magnification and higher resolution for this panel, since lipid-filled lysosomes are not clearly visible

Answer: Done as suggested.